# Management of Adult Patients with Gastrointestinal Symptoms from Food Hypersensitivity—Narrative Review

**DOI:** 10.3390/jcm11247326

**Published:** 2022-12-09

**Authors:** Alina Kanikowska, Susanne Janisz, Dorota Mańkowska-Wierzbicka, Marcin Gabryel, Agnieszka Dobrowolska, Piotr Eder

**Affiliations:** Department of Gastroenterology, Dietetics and Internal Diseases, Poznan University of Medical Sciences, Przybyszewski St. 49, 60-355 Poznan, Poland

**Keywords:** food allergy, food intolerance, gastrointestinal disorders

## Abstract

The incidence of food hypersensitivity has increased dramatically over the years not only among children but also in adults. Adult patients are usually less suspected of food hypersensitivity symptoms since food allergies are more typical for small children, with a tendency to outgrow the condition. The aim of this article is to increase awareness of hypersensitivity to food symptoms and their diagnosis and treatment possibilities among gastroenterologists and other health care professionals dealing with this type of patient. Symptoms of many gastrointestinal disorders, especially functional, may be driven by different types of mechanisms, and food intolerance or allergy should be considered as a potential cause. This article presents the current understanding of the epidemiology, diagnosis and treatment of immune- and non-immune-mediated food-induced diseases. Diagnosis of food hypersensitivity is based mainly on medical history, different types of sensitivity tests, e.g., hydrogen breath test, specific IgE (sIgE) serum concentration, tissue eosinophil count, skin tests and oral food challenges considered as a “gold standard” for food allergy. Elimination diet and pharmacologic treatment for allergy symptoms are first-line therapies. Eosinophilic gastrointestinal diseases are often caused by non-IgE-mediated food allergies, require endoscopic biopsy samples to confirm diagnosis and proper elimination diet often combined with steroids or proton pump inhibitor agents for treatment. Mast cell activation syndrome (MCAS) derives from pathologic reaction of mast cells with increased tryptase serum level as a marker. Symptoms may occur in the digestive, respiratory, skin, neurologic and cardiovascular system. Treatment is based on histamine type 1, type 2 (H1, H2) receptor antagonists and other mast cell stabilizing agents. Carbohydrate intolerances are the most common type of food hypersensitivity in adult patients, and an elimination diet is effective for reducing symptoms. Food additives hypersensitivity remains difficult to diagnose, but use of a diet low in chemical substances alleviates symptoms and helps to diagnose the triggering factors.

## 1. Introduction

Food allergy is a result of genetic predisposition and environmental factors. The rapidly growing number of patients with allergic diseases over recent years cannot be explained only by changes in genome sequencing as it can be attributed to epigenetic mechanisms triggered by environmental factors. Different factors occurring during pregnancy and the first years of life have a major impact on the immune system and further development of allergies. Childbirth delivery modes, such as Caesarian section, quality malnutrition, e.g., vitamin D insufficiency, reduced consumption of omega-3-polyunsaturated fatty acids and antioxidants, low gut microbial diversity, exposure to certain viral, fungal, microbial agents, endocrine disrupting chemicals, e.g., bisphenols, phthalates, tobacco smoking, traffic-related air pollutants, indoor air pollutants, e.g., dust mites, food antigens and antibiotic use, are examples of many environmental factors suspected of epigenetic changes leading to allergy development. In adult populations, increased use of antacids and obesity are correlated with increased incidence of allergic diseases [1,2].

One of the protective mechanisms that stimulates innate immunity and has a positive influence on gut microbiome is exposure to endotoxin, a cell-wall component of Gram-negative bacteria present in higher levels in farming and rural environments [3]. A healthy gut microbiome is associated with a higher level of short-chain fatty acids (SCFAs) in feces. SCFA may induce epigenetic mechanisms that regulate gene expression, leading to food tolerance through histone deacetylase (HDAC) inhibition, influence on intestine wall integrity and immune cells engaged in the tolerance network [4,5]. In the Canadian Healthy Infant Longitudinal Development cohort, children with higher gut microbial richness at age 3 months were associated with 55% reduced risk for food sensitization by age 1 year. In food-sensitized infants at 3 months and 1 year in gut composition, *Bacteroidaceae* was under-represented, with over-presence of *Enterobacteria* compared to healthy controls [6]. Several studies indicate that certain strains of gut microbiota, e.g., *Clostridia* and *Bacteroidales* in particular, protect against food allergy through regulatory T cells influence [7]. Another protective mechanism against food allergy may be early oral introduction of allergen, as demonstrated in the clinical trial Learning Early about Peanut Allergy (LEAP) [8]. Epigenetic changes can be inherited and are induced by methylation of DNA to a cytosine in CpG dinucleotide sequences, histones alterations by different mechanisms (mainly methylation, acetylation, phosphorylation and ubiquitination) and involvement of micro-RNA in post-transcriptional gene regulation [9,10,11].

Food hypersensitivity is thought to be a problem in infants or younger children, with a tendency to acquire tolerance over the years. In adults, usually, lactose intolerance, often confused with cow milk allergy, is widely known and included in differential diagnosis of gastrointestinal diseases. However, milk sugar intolerance is only one of many possible factors responsible for diarrhea and abdominal bloating [12].

Due to the growing number of food allergies and intolerances, an increasing number of children and adult patients search for help by addressing gastroenterologists and other health care providers. These symptoms are often treated for years as functional gastrointestinal disorders (FGD), such as irritable bowel syndrome (IBS) or gastroesophageal reflux disease, are treated with pharmaceutical agents that alleviate symptoms while the root cause of the problem remains unrecognized [13]. Since food hypersensitivity is usually treated with an elimination diet, it is essential to diagnose the disturbing food element. For the gastroenterologist, it is important to know which signs should be connected with a possible food hypersensitivity background and which diagnostic tools can be used.

The aim of this article is to increase awareness of hypersensitivity to food symptoms, their diagnosis and treatment possibilities in adult patients since this group is rarely suspected of food allergies usually assigned to children. Celiac disease, an autoimmune disease elicited by gluten ingestion, will not be included in this review as it is a well-known entity in the gastroenterology field.

According to the World Allergy Organization, food hypersensitivity is when the dose tolerated by normal subjects causes reproducible symptoms or adverse reactions [14]. Food hypersensitivity may be divided into immune-mediated—food allergy with immunological response—and non-immune-mediated—food intolerance without immunologic mechanisms involved, where the reactions may be due to a lack of metabolizing enzymes, toxic or pharmacological factors [15]. The prevalence of food allergy is estimated to be 6–10% for children and 2–5% for adults [1,16].

## 2. Methods

A PubMed and Google Scholar literature search for the period between 2004 and October 2022 was performed using the terms and variants of food hypersensitivity, IgE, mixed and non-IgE food allergy, food intolerance, eosinophilic gastrointestinal disorders, mast cell activation syndrome, carbohydrate intolerance, non-celiac gluten sensitivity and additives hypersensitivity. Manuscripts were selected from randomized controlled trials, reviews, systematic reviews, meta-analysis, consensus guidelines, prospective and observational studies. Single-case reports, abstracts/posters, articles with duplicated data, articles in non-English language were excluded. We excluded also articles concerning celiac disease and typical pediatric manifestations of food allergy, e.g., food-protein-induced enterocolitis and proctocolitis. Controlled vocabulary and keywords associated with each concept were examined and combined with Boolean operators in a logical way. References from relevant papers were also searched for other appropriate studies and included in the review.

## 3. Food Allergy

### 3.1. Pathophysiology

In healthy circumstances, food antigens crossing the intestinal mucosal barrier elicit a tolerance response from macrophages, CX3CR1^+^ antigen presenting cells (APCs) and CD103^+^ dendritic cells (DC), resulting in differentiation of T naive cells into T regulatory (Treg) cells producing IL-10 and an increase in IgG4 and IgA production by B cells. In food allergy antigen presenting cells (APCs), mainly DC, they activate T naïve cells differentiation into T helper cell 2 (Th2), which induces B-cells-specific IgE (sIgE) production to food antigen and, through proinflammatory cytokines, activates eosinophils, mast cells and basophils responsible for allergy symptoms [17]. While it is still not fully clear what mechanisms lead to tolerance break, it is observed that several tissue-derived cytokines, e.g., IL-33, probably through activation of innate lymphoid cells (ILCs), lead to Th2-biased immune responses to food along with exposure to certain microbial toxins, food components, defective barrier functions and microbiome dysbiosis [18].

Allergy to food may be IgE-mediated, mixed: IgE and non-IgE-mediated or non-IgE-mediated. IgE-mediated allergy is a type I hypersensitivity reaction. Antigen food-specific IgE binds to the FcεRI receptors on mast cells and basophils and degranulates mediators, such as histamine, tryptase, platelet activating factor, prostaglandins and leukotrienes, that cause typical allergy symptoms, such as rash, pruritus, edema, bronchoconstriction and vasodilation [1,15].

Non-IgE-mediated allergies involve antigen-specific T cells, cytokines, mast cells and eosinophils but not IgE production, leading to tissue inflammation, e.g., food-protein-induced enterocolitis syndrome (FPIES), food-protein-induced allergic proctocolitis (FPIAP) and food-protein enteropathy (FPE), which mainly develop in infants, rarely in adults [19]. Eosinophilic gastrointestinal disorders (EGIDs) may be present in children and in adults, and, sometimes, IgE mechanisms may be involved, as in mixed IgE and non-IgE-mediated allergy, where both immune mechanisms may be responsible for symptoms development [13].

### 3.2. Diagnostic Tools for Food Allergy

While taking the history of the patients, it is important to ask about allergies. Usually, patients are aware of allergies to drugs or aeroallergens causing symptoms in the respiratory tract but rarely realize that an allergy may be the culprit of their gastrointestinal problem. If one type of food elicits sudden symptoms of mouth swelling, urticaria or pruritus after each consumption, then a food allergy diagnosis is usually easy [14]. The problem starts when the symptoms are not obvious and may imitate other frequent diseases. Symptoms that should always raise suspicion of allergy are angioedema of the lips, tongue or pharynx, oral pruritus, persistent gastroesophageal reflux, loose or frequent stools, diarrhea, blood, mucus in the stool, abdominal pain, dysphagia, food refusal or aversion, constipation, perianal redness and pruritus, pallor and tiredness, nausea, vomiting or recurring urticaria [2].

Carefully taken anamnesis is the most important step in the diagnosis. Suspicion of food allergy should always involve detailed clinical and dietary history as the most important diagnostic tool. Awareness should be focused on reproducible reactions after food ingestion, contact or even inhalation, presence of co-factors such as alcohol consumption, use of nonsteroidal anti-inflammatory drugs, exercise, menstruation or infection, which may facilitate an allergic reaction. Typically, IgE-mediated reactions occur within minutes to an hour, while non-IgE-mediated reactions may also develop several hours later. A food diary correlating with the symptoms is very helpful to decide what type of hypersensitivity reaction patients suffer from [20].

To confirm the suspected allergen, different tests may be conducted. Non-IgE-mediated allergy is not easy to diagnose. Typically, elimination diet with oral food challenge (OFC) or double-blind placebo-controlled food challenge (DBPCFC) and histopathological evaluation are needed, as in, e.g., eosinophilic esophagitis. Both OFC and DBPCFC should be conducted by experienced personnel in case severe anaphylactic reactions appear. In non-IgE-mediated allergies, a diagnostic food elimination diet usually lasts 6 weeks [21].

IgE-mediated reactions can be verified by sIgE measurement from blood and/or skin prick tests (SPT) with standardized extracts and/or with fresh food in a procedure called “prick to prick”, where the lancet scratches the fresh food and then skin. Positive results for SPT are based on the mean wheal diameter ≥ 3 mm and for sIgE ≥ 0.35 kU/L, and they show sensitization to allergens. It is assessed that approximately half of sensitized individuals are able to tolerate the food without any symptoms [20].

The sensitivity and specificity for SPTs vary based on several factors, e.g., age, prevalence of food allergy in studied group, type of tested allergen kit and testing technique. Usually, the larger the skin wheal in SPT or the higher the sIgE level, the higher the likelihood of clinical reactivity to the tested allergen. Sampson et al. assessed in DBPCFC in a group of children with skin allergy the predictive indices for SPT and sIgE assessed by radio-allergosorbent test (RAST). SPTs and RAST had good negative predictive accuracy indices (82% to 100%) but poor positive predictive value, for SPTs 25% to 75% and 0% to 57% for RAST [22]. Considering other studies, it is estimated that sIgE sensitivity is 10% to 25% for clinically relevant reactions and depends on the diagnostic method [23].

It is important to test only those suspected allergens that were selected through careful anamnesis. Otherwise, unnecessary elimination diets may be implemented according to positive SPT and/or RAST. In vitro tests should be performed in patients without possibility to stop antihistaminic drugs, contraindicated 10–14 days prior to SPT, with severe atopic dermatitis or in case of possible severe anaphylactic reaction with tested allergens. Test results should be confirmed by elimination diet at least 2–4 weeks and OFC or DBPCF. The gold standard diagnostic test for all types of food allergy is DBPCF [1,23].

More precise in vitro tests are based on component-resolved diagnostics (CRD), where antibody against a specific component of complex allergen is tested, which allows to predict the seriousness of an allergic reaction; e.g., allergy to peanut with an increased level of antibodies against Ara h 8 (PR -10 protein) is usually confined to oral allergy syndrome and provokes mild reactions, while antibodies against Ara h 2 (storage protein) are responsible for anaphylactic shock [23,24].

Other tests that may predict the severity of the reaction are also an in vitro test based on flow cytometry called the basophil activation test (BAT), measuring cell surface expression of CD63 and CD203c on exposure to allergen, or mast cell activation test (MAT), measuring cell surface expression of CD63 and CD107a, but its availability is scarce [23,25].

Not recommended tests for food allergy diagnostics are the measurement of sIgG, sIgG4, bioresonance therapy, electrodermal testing and hair analysis [26].

### 3.3. Food Allergy Treatment

Treatment of food allergy is based mainly on an elimination diet of the harmful allergen and the use of antihistaminic drugs, such H1- and H2-receptor antagonists, anti-leukotriene drugs or mast cells (MC) stabilizing agents; in case of anaphylactic shock, possibility epinephrine self-injectors, at least two in case of prolonged reaction, should be prescribed. All patients should receive an emergency action plan with written instructions on treatment guidelines and food triggering factors list to avoid. Dietician consultation with education in food labeling is a necessary step in proper adherence to the elimination diet [27].

The use of probiotics and influence on the gut microbiome is another possible way of allergy treatment. Thus far, *Lactobacillus rhamnosus GG* is considered to induce tolerance to milk in children with cow milk allergy [28]. Studies conducted on prevention and treatment of food allergy with probiotics are often contradictory, probably because of different strains of bacteria, dose and therapy duration [29]. Consumption of baked milk and egg may enhance allergy resolution, and tolerance of such processed food allergens may be a sign of immunologic tolerance development [27].

Immunotherapy, effective in pollen allergy, has mainly desensitizing effect in food allergy. It requires daily administration of food protein, and sustained unresponsiveness after discontinuation is variable. Food immunotherapy is dedicated for Ig-E mediated allergy and is mainly used to increase the amount of accidentally ingested allergen as a protection from anaphylactic shock. There are different forms of immunotherapy studied: oral, sublingual and epicutaneous, usually with heterogeneous protocols and without established standardization. Meta-analyses of twenty-five trials investigating oral immunotherapy, five studies sublingual immunotherapy and one study evaluating epicutaneous immunotherapy demonstrated significant positive results in terms of desensitization (risk ratio (RR) = 0.16, 95% CI 0.10, 0.26) and suggested sustained unresponsiveness (RR = 0.29, 95% CI 0.08, 1.13) [30]. Research studies include also use of omalizumab as a modulatory agent enhancing immunotherapy efficacy [31,32]. There is a new drug approved for oral immunotherapy used in allergy to peanuts in children that aims to increase tolerance of the allergen and provides protection against accidental exposure. The drug, approved by the FDA, is called Palforzia (peanut allergen powder), and increased doses are given in healthcare settings [33].

## 4. Clinical Manifestation of Food Allergy in Adults

### 4.1. IgE-Mediated Food Allergy

Allergy in children and adults may be elicited towards all types of food content. Typical allergic symptoms from the gastrointestinal tract are abdominal pain, vomiting, diarrhea, bloating, heartburn, constipation; from skin—urticaria, edema, flush, pruritus, atopic dermatitis exacerbation—and from respiratory tract—rhinitis, asthma exacerbation [1,2]. The prevalence of specific food allergens differs between these groups. In children, typical allergens are the so called “big eight”: milk, eggs, fish, crustacean/shellfish, tree nuts, peanuts, wheat and soy beans [24]. Allergy to milk, egg, wheat and soy has a tendency to resolve with age, but peanut, tree nut, seeds, fish and shellfish allergies usually persist through adult life. For adult patients with aeroallergy, cross-reactions with food are possible due to similar proteins in food and pollen to which the same antibody reacts. It may be an allergy to one or multiple foods (Table 1).

The most common allergic reaction in adults is oral allergy syndrome (OAS), characterized by pruritus and/or edema of the oral cavity, sometimes throat after ingestion of certain foods—usually raw fruits, vegetables or nuts—as a cross-reaction to pollen allergy. The symptoms are usually mild and disappear within a couple of minutes and rarely lead to anaphylaxis. Depending on the type of allergens, there are factors such as high temperature, pasteurization and food processing that usually decrease, but, for a few allergens, they increase allergenic potential due to conformational changes. Profilins are sensitive to heat and gastric acid, so allergic reaction is caused only by raw fruits and vegetables and confined to the oral cavity.

Non-specific lipid transfer proteins (nsLTP) are insensitive to heat and proteolysis with the highest concentration in the peel of the fruit and may be responsible for more severe symptoms, including anaphylactic shock, after consumption of raw or cooked food. NsLTP is usually responsible for allergy to nuts and typically for patients from the Mediterranean area to peach, apricot and plums [23,24]

The so called “pancake syndrome” (oral mite anaphylaxis) may occur in atopic patients allergic to mites after mite-contaminated flower food ingestion. Typical symptoms are breathlessness, angioedema, wheezing and rhinorrhea [35].

Treatment is based on an elimination diet—in the majority of cases, food processing as heating changes the conformation of the allergen—and use of antihistamines and adrenaline injector [36,37].

There is an unusual type-1 allergic reaction with IgE against galactosyl-α-(1,3)-galactose (α-Gal) that is delayed to 3–6 h after ingestion of red meat or innards. This type of sensitization is developed by a tick bite. Cetuximab or rarely infliximab treatment may elicit an allergic response in this group of patients. It is advised to use nonglycosylated monoclonal antibodies in such cases [38].

FDEIA—food-dependent exercise-induced anaphylaxis—is an allergic reaction after consumption of food, e.g., wheat, especially omega-5-gliadin, shellfish, nuts and many others, that occurs up to four to six hours after physical exertion. If the consumed food is not followed by exercise, it does not elicit an allergic reaction. Symptoms may vary from mild, such as urticaria, pruritus, headache, profuse sweating, dysphagia, abdominal pain or diarrhea, to the most serious reaction, such as anaphylactic shock. Treatment requires elimination diet up to 6 h before exercise, use of H1-blokers and adrenaline injector if needed [39].

### 4.2. Non-IgE-Mediated or Mixed Allergy

#### 4.2.1. Eosinophilic Inflammation of the Digestive Tract

Eosinophilic esophagitis (EE), gastroenteritis (GE) or colitis (EC) are characterized by eosinophilic infiltration of the digestive tract wall. Depending on which layer is inflamed, symptoms may present differently, but the leading symptom is abdominal pain, dysphagia and food impaction (Table 2).

EE may be present among up to 8% of patients with gastroesophageal reflux disease that does not respond to proton pump inhibitors (PPI). EE is more prevalent in patients with connective tissue disorders with hypermobility, such as hypermobile type Ehlers–Danlos syndrome (hEDS) or hypermobile spectrum disorder (HSD) [41]

Pathophysiology

The most probable cause is hypersensitivity to food, more often in the non-IgE-dependent mechanism, but eosinophilic response in the digestive tract can be triggered also by aeroallergens. Interleukin-5 is the most important cytokine to attract eosinophils.

Other substances such as eotaxin-1 and α4b7 integrin also take part in the inflammatory process. Patients with eosinophilic inflammation usually suffer from other allergic diseases, such as pollen rhinoconjunctivitis, peripheral blood hypereosinophilia and elevated IgE level [40,42].

Diagnosis

The presence of symptoms along with endoscopic investigation with biopsy samples is essential for diagnosis of EE. None of the symptoms are pathognomonic, but their presence may indicate EE. Endoscopy may reveal linear furrows, concentric rings in the form of “trachealization”, mucosal fragility, strictures, narrow caliber esophagus, white exudates or decreased vascularity. Around 10% of patients with dysphagia due to EE have no visual changes of the esophageal mucosa according to the Prasad et al. study, which means it is important to take biopsies in suspected cases even from healthy-looking mucosa [42,43]. To increase sensitivity, it is important to take 6 samples from both the proximal and distal parts of the esophagus [44]. Histopathological examination typical for eosinophilic inflammation of the digestive tract requires for eosinophil count at least 15 eosinophils per high-power microscopy field (HPF) for the esophageal biopsy and more for other parts of digestive tract (Table 3).

It is important to exclude other causes of eosinophilia, such as a parasitic infestation or drug hypersensitivity.

Treatment

A proper elimination diet is the safest way for induction and maintenance therapy. There are three types of elimination diets used in treatment: the most popular empiric diet, IgE-based diet, and elemental diet. The most effective is an elemental diet based on aminoacid formula, but, due to its taste and cost, it can be used for a short period of time. When elemental diet leads to healing of esophageal mucosa, gradually, different groups of food allergens are added to the diet. After each food group, endoscopy should be performed to check eosinophil count [44]. The next option is an elimination diet based on IgE tests—sIgE or/and SPT—but the detected food allergens are usually not alone to develop inflammation because the main pathophysiological reaction is non-IgE-dependent, often with delayed response [48]. The most popular option is to implement an empiric diet with the elimination of six foods: wheat, milk, eggs, nuts/peanut, soy and seafood, without any allergen testing. In this case, histological remission is observed in 70–80% of patients. Since the 6 foods elimination diet is difficult to follow, it is advised to start with step-up approach—first with the elimination of most typical allergens for EE: milk and wheat, and, if it is not sufficient, the rest of the allergens should be eliminated [44,49].

Pharmacologic therapy is reserved as a last resort or for patients with intense, requiring fast withdrawal symptoms, abd also if the diet treatment is ineffective or not followed. First PPI—twice daily for 8 weeks—should be used. PPI suppresses Th2-associated cytokine expression and reduces eosinophilic inflammation. If the symptoms or endoscopic results are not improved, glucocorticoids should be introduced [50]. Glucocorticoids are most effective in decreasing the number of eosinophils by inducing their apoptosis. Preferably, first-line treatment is topical steroid therapy. Fluticasone 440–880 μg from multi-dose inhaler swallowed not inhaled or 1–2 mg budesonide in orodispersible tablets or aerosol or suspension, twice daily for 8 weeks, are used. For small children, a viscous type of budesonide is recommended, with sucralose added to improve the taste. For severe cases and patients that do not respond to topical therapy, prednisone 0.5–1 mg/kg for two weeks is recommended. In 6–8 weeks, the dose should be tapered. After taking oral glucocorticoids, it is recommended to rinse out the mouth to prevent oral candidiasis and avoid eating for 30 min to let the drug act on the mucosa. In case of esophageal stricture or narrowing, endoscopic esophagus dilations should be performed [51,52,53].

In GE and EC, first-line treatment to induce remission is oral prednisone 0.5–1 mg/kg/d for 6–8 weeks, with tapering dosage. If symptoms are still present after withdrawal, prednisone 5–10 mg/d or budesonide 3–9 mg should be taken to sustain remission. The elimination diet is not studied in adults, and it is not known if it is as effective as in EE [46,47].

Mesalazine is effective in diminishing symptoms in EC. Montelukast–anti-leukotriene D4 receptor antagonist is more effective in patients with EGE or EC and not effective in EE. In severe steroid-resistant cases, azathioprine may be effective [54]. Biologic therapy available for other allergic disease treatment, such as anti-IL-5 monoclonal antibodies–mepolizumab, anti-IL-4r blocker dupilumab or omalizumab–anti-IgE monoclonal antibody, were studied in the context of EE treatment, but only dupilumab seems to have sufficient efficacy, while omalizumab and mepolizumab decreased the count of eosinophils in mucosa but did not alleviate symptoms [55,56,57].

#### 4.2.2. Local Immune Response to Food Antigens

Use of confocal laser endomicroscopy in the study by Fritscher-Ravens et al. analyzing duodenal tissues’ responses to food components from patients with IBS showed increased eosinophil activation and disruption of the intestinal barrier mediated by nonclassical food allergy [58]. New concept of local food allergy has been proposed by Aguilera-Lizarraga et al. based on observation in animal and human models that, in response to food intake, increased mast cells accumulation (MC) and release of MC-mediators stimulate visceral hypersensitivity, considered the main pathomechanism in IBS. The observed response in contrast to food allergy was local, and IgE-specific antibodies were detected only in colonic tissue, which would explain negative screening for food-specific IgE in blood and skin and difficulties in finding a food allergen [59].

Mast cell dysfunction plays also an important role in triggering symptoms of food adverse reaction in other diseases. Food allergy may be a part of mast cell activation syndrome and should be considered in differential diagnosis.

## 5. Mast Cell Activation Syndrome (MCAS)

Mast cells (MC) are immune cells present in tissues responsible for many protective functions in the body, taking part in inflammation against bacteria, virus, fungi and parasites, but they are also responsible for initiation and retaining allergic response and anaphylactic reaction. They produce pro-inflammatory mediators, such as tryptase, chymase, histamine, cysteinyl leukotrienes and prostaglandins, but also heparin and other proteoglycans by inducing a variety of receptors on mast cells, such as Toll-like receptor (TLR), complement receptor (CR) and immunoglobulin receptors IgR. The level of intensity of the allergic reaction depends on many factors, such as co-existence of triggering factors, e.g., exercise, alcohol, drugs, hormones, temperature. MC play an important role in GI tract, regulating the endothelial and epithelial function of the intestine, such as preserving epithelial barrier integrity, influence on endothelial permeability, neuronal function influencing pain mediation and peristalsis, wound healing fibrosis, and immune cells migration. Importantly, MC are influenced by both acute and chronic stress, they release pro-inflammatory mediators and motor neuron response is diarrhea, cramps and abdominal pain. MC are essential in gut homeostasis and, with abnormal function, are thought to play an important role in the pathogenesis of eosinophilic inflammation of the digestive tract, irritable bowel syndrome, dyspepsia, appendicitis, inflammatory bowel disease, celiac disease, Hirschsprung’s disease and intestinal neuronal dysplasia [60,61]. Basal serum tryptase level is constant in healthy individuals, but, after the anaphylactic reaction, the level rises substantially, and this difference in levels is important for diagnosis of MC activation.

MCAS classifies chronic pathological reactions of MC in three subtypes: primary, secondary and idiopathic depending on the detection of D816V mutation in C Kit gene (Table 4).

It is often more frequent in people with postural orthostatic tachycardia syndrome (POTS) and hEDS. It was also observed that small intestinal bacterial overgrowth (SIBO) often coexists with MCAS [61,62].

**Table 4 jcm-11-07326-t004:** Subtypes of mast cell activation syndrome (MCAS) [61,62,63,64].

Subtype of MCAS	Diagnostic Criteria
Primary	Detectable D816 V mutation in C Kit gene, with MC CD25 expresssion, with or without underlying mastocytosis
Secondary	Underlying background, such as allergic diseases, e.g., food allergy, autoimmune diseases, infections, neoplasms
Idiopathic	If both subtypes are excluded

MCAS—mast cell activation syndrome, MC—mast cell, CD—cluster of differentiation.

MCAS is a chronic condition caused by abnormal function of MC that may be manifested by a variety of symptoms affecting different systems and organs (Table 5).

There is also a possibility of an autosomal dominant condition called hereditary alpha-tryptasemia (HAT), caused by replications of the *TPSAB1* gene encoding alpha-tryptase. Individuals with this mutation tend to have higher concentration of tryptase in the blood, usually higher than 8 ng/mL as a baseline, and often present the same spectrum of symptoms as other patients with MCAS [63,64].

### 5.1. MCAS Diagnosis

According to P. Valent et al., MCAS may be diagnosed if:there is the presence of typical clinical symptoms from at least two systems (respiratory, skin, digestive, neurologic, cardiovascular), usually with episodes of anaphylaxis,there is an increase in the serum tryptase level above the individual’s baseline serum tryptase (sBT), counted according to formula 20% + 2 (=sBT × 1.2 + 2), and/or increase in other MC mediators in biological fluids (histamine and its metabolites, prostaglandin D2). It is important to have a minimum of 2 measurements of tryptase level, preferentially one from the asymptomatic period and the other after the anaphylactic episode, one hour after if possible.there is a response to drugs targeting MC activation, mediator release from MCs and/or MC mediator effects by the partial or total withdrawal of symptoms.

There are also patients with MC activation that do not fulfill the criteria of MCAS but suffer from symptoms localized to one system and/or without anaphylactic reactions [65].

It is postulated that, due to a lack of good diagnostic tools, both MCAS and histamine intolerance (HIT) may be co-existing or misdiagnosed, so it is important to take both of these entities under consideration [66].

### 5.2. MCAS Treatment

Treatment: MC activation may be decreased with H1-receptor antagonists, H2-receptor antagonists, anti-leukotriene medications or MC stabilizing agents. MCAS patients are also advised to carry two (or more) epinephrine self-injectors (Table 6). A low-histamine diet and elimination diet including detected food allergens is also recommended since some foods may trigger mast cells activation [64,65].

## 6. Non-Immune-Mediated Adverse Food Reactions (Food Intolerances)

### 6.1. Carbohydrate Intolerance

Food intolerances are adverse reactions that do not involve the immune system. They may be caused by pharmacologic mechanisms; e.g., caffeine has different pharmaco-kinetics depending on *CYP1A2* genetic variant, similar to alcohol, depending on alcohol dehydrogenase (ADH) aldehyde dehydrogenase (ALDH) genetic variants, toxic reactions, e.g., high load of histamine in rotten fish or metabolic reactions, e.g., lactase deficiency. In adults, non-immune-mediated food reactions are more common, often caused by carbohydrate intolerance (Table 7).

#### 6.1.1. Lactose

Lactose is a disaccharide that is digested in the mid-jejunum on the brush border of the villi. The alpha-glucosidase activity cleaves lactose into monosaccharides glucose and galactose. Primary lactase deficiency is most common and caused by declining lactase activity with age, depending on genetic factors. Lactase is located at the end of the intestinal villi and is susceptible to injury, leading to deficiency in the patients, what is termed as secondary lactase deficiency, and it can be reversible if the villi are restored. This type of lactase deficiency is often observed in celiac disease, gastroenteritis and inflammatory bowel diseases. There are also other forms, such as congenital lactase deficiency, due to autosomal recessive gene mutation, a rare but severe form of intolerance leading to malnutrition since it occurs in newborns and lactose is the main sugar in human milk. Developmental lactase deficiency is typical for premature infants and is caused by immature intestine cells, which improves with time and intestine maturation. When undigested lactose comes in contact with the intestinal microbiota as a consequence of bacterial fermentation, the production of gasses, such as hydrogen, carbon dioxide and methane, and short-chain fatty acids occurs, resulting in intolerance symptoms. Typical symptoms after ingestion of lactose are abdominal pain, bloating, borborygmi or diarrhea. The symptoms vary by the amount of lactose ingested with each person, and it is considered multifactorial [12,69].

Lactose intolerance can be tested during a hydrogen breath test. Usually, 25g of lactose is ingested, and breath hydrogen level is observed up to 3 h. If the level rises > 20 ppm compared with baseline, the test is positive, but first SIBO should be excluded [70]. Stool pH can be assessed as unabsorbed lactose ferments into lactic acid. Other possibilities for secondary lactase deficiency are small bowel biopsy and primary lactase deficiency testing for gene mutations LCT-13910C/C and LCT-22018G/G [70,71]. The aim of lactose intolerance treatment is to reduce lactose amount in the diet to a level that is well-tolerated. This can be increased with the appropriate microbiota support, such as *Lactobacillus* and *Bifidobacterium*, with the assistance of prebiotics, especially galactooligosaccharides (GOS). Additional treatment using lactase supplementation can improve digestion and reduce symptoms [72].

#### 6.1.2. Fructose

After lactose, fructose is the second most frequent carbohydrate to cause adverse effects in the digestive tract. Fructose is absorbed from the small intestine by facilitated diffusion, mainly through GLUT-5 transporter expressed on the apical membrane, with the glucose enhancement. It is usually free fructose or total high fructose content that, by osmotic mechanism, causes fluid influx and gas production in the intestine lumen. Diagnosis may be established based on a hydrogen breath test after ingestion of 25g fructose solution, but data show that results and better diagnosis are based on questionnaires, such as self-reported dietary fructose intolerance (DFI) [73,74]. Treatment is based on reducing the amount of fructose in diet or choosing foods with a balanced ratio of fructose to glucose level, preferably 1:1, which enhances fructose absorption. Patients usually can tolerate 10–15 g of fructose per day [75].

#### 6.1.3. FODMAPs

Lactose, fructose, fructans, fructooligosaccharides (FOS), galactooligosaccharides (GOS) and polyols, such as sorbitol, mannitol, maltitol, xylitol, polydextrose and isomalt malabsorption, are often a cause of chronic diarrhea, bloating and flatulence in patients with irritable bowel syndrome [76]. The most effective treatment for patients with food intolerance to various nutrients is a diet low in fermentable sugars, called a low-FODMAP diet, an acronym for fermentable oligo-, di-, monosaccharides and polyols. Symptoms are reassessed after 6–8 weeks and compared to pre-diet clinical symptoms [77,78,79].

#### 6.1.4. Non-Celiac Gluten Sensitivity (NCGS)

Another non-immune-mediated type of reaction is non-celiac gluten sensitivity (NCGS) or non-celiac wheat sensitivity (NCWS).

Celiac disease and wheat allergy involving pathologic reactions to gluten are well-known diseases with approved diagnostic guidelines. NCGS should be suspected in patients with both of them excluded; unfortunately, so far, there are no sensitive and reproducible biomarkers that would allow to make the diagnosis. It is suggested that gluten is not the only trigger of symptoms as amylase-trypsin inhibitors (ATIs) and FODMAPs may also be responsible for eliciting symptoms. The only accessible recommendations for diagnosis of NCGS were created by the Salerno Experts in 2014. Confirmation of the diagnosis includes assessing the clinical response to the gluten-free diet and the effect of reintroducing gluten after a period of treatment. Clinical evaluation is performed using a modified version of the Gastrointestinal Symptom Rating Scale (GSRS), evaluating common symptoms of gastrointestinal diseases [80]. Typical symptoms of NCGS are bloating, lack of wellbeing, abdominal pain, tiredness, diarrhea, headache, epigastric pain, anxiety, nausea, foggy mind, joint and muscle pain, aphthous stomatitis and skin rash/dermatitis. HLA-DQ2/DQ8 haplotypes and anti-gliadin antibodies IgG (IgG-AGA) are found in approximately 50% of NCGS patients. Treatment is gluten or wheat elimination diet, or, in some cases, low FODMAPs diet [81].

## 7. Mixed Type of Reactions

While many different types of food reactions may occur, hypersensitivity to additives includes immune and non-immune-mediated types of reactions.

The food industry uses food additives abundantly for food preservation and to improve food’s taste or color. Foods also have natural chemical compounds that may trigger intolerance symptoms. In some cases, IgE reaction is possibly involved. Adverse reactions described after ingestion of food additives are flushing, urticarial, angioedema, rhinorrhea, abdominal pain, diarrhea, depressed mood and fatigue. Higher adverse reaction frequency is observed among patients with allergy, especially asthma [82].

The most typical additives responsible for hypersensitivity are sulfites, benzoates, monosodium glutamate, salicylates and colorants [83,84]. Sulfites are metabolized by sulfite oxidase. Insufficiency of this enzyme predisposes to intolerance symptoms. Sulfites are present in substantial amounts in, e.g., cider, dried fruits and white wine, lettuce, dehydrated potato, shrimp, prawn and lobster. Benzoates are synthesized by plants and present in animal products, such as dairy and milk, cranberries and prawns, but also in higher concentrations are added to processed food, such as sweets, soft drinks and ice creams. Salicylates are present in fruits and vegetables, herbs and spices, and hypersensitivity symptoms include usually asthma, rhinosinusitis and urticaria, rarely digestive tract [85].

All these additives may elicit symptoms from the skin, respiratory and digestive tract in hypersensitive patients. Diagnosis is based on OFC but often is not helpful since the symptoms may be present only after combined exposure and delayed in time [86,87]. Food chemical elimination diet, e.g., Royal Prince Alfred Hospital (RPAH), diet is worth trying, especially in patients with chronic diarrhea without evident cause, patients with urticaria, asthma or rhinosinusitis. RPAH elimination diet includes three restriction levels: strict, moderate and simple depending on symptoms magnitude and personal preference. It requires avoidance of naturally occurring or artificially added substances, such as salicylates, amines, monosodium glutamate, preservatives benzoates, propionate, sulfites, nitrites, sorbic acid, colorings, flavorings and, in some cases, gluten, dairy and soy. Assessment of clinical improvement eliminated substances challenges with subsequent diet liberalization and allows to establish the accepted amount of chemical intake for each patient without triggering symptoms [88,89].

The selection of discussed hypersensitivity syndromes was influenced by clinical practice observations and, therefore, presented as a narrative review. We have mainly focused on the gastrointestinal symptoms from food hypersensitivity, i.e., reaction types approved by international medical societies, omitting less approved or not evidence-based syndromes, such as histamine intolerance or systemic nickel allergy syndrome. The limitations of the presented data result from the searching strategy adopted in this paper, which is not designed as a systematic review, with a limited number of databases for the identification of other potentially eligible studies. We were unable to investigate in detail the role of many study design characteristics, such as adjustment for potential confounders, follow-up length, age range or other factors that may play a role in the final study outcome and conclusions. Another limitation is that we have only included studies written in English, and, therefore, original works published in other languages are excluded.

## 8. Summary

There has been a growing number of different types of hyperactivity reactions to food discovered that may be mimicking or underlying development of FGD, e.g., IBS, chronic diarrhea, dyspepsia or other gastrointestinal disorders [90]. Better knowledge and diagnosis of these mechanisms, their symptoms and specific, targeted treatment lead to better care of patients. Correct diagnosis often requires not only an endoscopic examination, blood and stool tests but also tools detecting allergic and intolerance causes, such as skin prick tests, sIgE concentration, CRD, BAT, MAT, hydrogen breath tests or genetic testing. There are many challenges in the food hypersensitivity diagnostic field. Some possible mechanisms, such as local immune response to food allergens, are still unavailable for routine assessment. Triggering factors for eosinophilic gastrointestinal diseases involving non-IgE and mixed type of allergic reactions are often difficult for in vitro screening. Elimination diet or pharmacologic treatment effectiveness requires systematic endoscopies with tissue sampling. Use of invasive procedure as a control diagnostic tool often leads to underdiagnosis and undertreatment of eosinophilic inflammation in the digestive tract. Food hypersensitivity included in broad spectrum symptoms of other immune disorders is rarely considered in differential diagnosis, e.g., MCAS. Knowledge regarding possible hypersensitivity food reactions is essential in the general practice of different health professionals.

## Figures and Tables

**Table 1 jcm-11-07326-t001:** Selected cross-reactions between allergens [23,34].

Allergen	Cross-Reactive Food Allergen
Birch pollen	Apple, peach, cherry, kiwi, peanut, almond, hazelnut, carrot, celery, soy
Grass pollen	Peach, orange, celery, tomato, melon
Ragweed pollen	Celery, carrot, parsley, onion, leek, paprika, pepper, chamomile, mustard, sunflower, cabbage, cauliflower, corn
Cat’s fur	Pork
Mites and cockroach	Crustacean shellfish
Latex	Banana, avocado, kiwi, chestnut, potato, papaya

**Table 2 jcm-11-07326-t002:** Symptoms depending on eosinophilic esophagitis (EE) inflammation grade [40].

Digestive Tract Layer	Symptoms
Mucosa	Dysphagia, heartburn, lack of appetite, food impaction (often meat), diarrhea, vomiting, symptoms of malabsorption
Muscular layer	Additional wall thickening, stricture, volvulus, intussusception, perforation
Serosal layer	Ascites with fluid and a high count of eosinophils, peritonitis in more severe cases

**Table 3 jcm-11-07326-t003:** Eosinophilic count per high-power microscopy field (HPF) in different types of eosinophilic inflammation [45,46,47].

Type of Eosinophilic Inflammation	Eosinophilic Count per HPF
EE	≥15 eosinophils
GE	≥20 eosinophils
EC	≥50 eosinophils in the right colon ≥35 eosinophils in the transverse colon ≥25 eosinophils in the left colon

EE—eosinophilic esophagitis, GE—eosinophilic gastroenteritis, EC—eosinophilic colitis, HPF—high-power microscopy field.

**Table 5 jcm-11-07326-t005:** Symptoms of mast cell activation syndrome (MCAS) [62,63,64].

Location of MCAS	Symptoms
Gastrointestinal system	Abdominal pain, bloating, constipation, diarrhea, nausea, gastric hypersecretion, dyspepsia, heartburn
Skin	Urticaria, flushing, pruritus, angioedema
Respiratory	Wheezing, throat swelling, hoarseness, nasal congestion, nasal pruritus
Neurologic	Headaches
Cardiovascular	Hypotensive syncope or near syncope, tachycardia, anaphylactic shock

**Table 6 jcm-11-07326-t006:** Medical treatment of mast cell activation syndrome (MCAS) [64,65].

Class of Medication	Examples
H1-receptor antagonist	Cetirizine, Bilastine, Desloratadine, Fexofenadine, Rupatadine
H2-receptor antagonists	Famotidine, Ranitidine
Anti-leukotriene medications	Montelukast
MC stabilizing agents	Ketotifen, Cromglycean
Sympathomimetic catecholamine	Epinephrine

MC—mast cell, H1,H2—histamine type 1, type 2.

**Table 7 jcm-11-07326-t007:** Carbohydrate intolerances: diagnostic tools and treatment [67,68].

Type of Intolerance	Diagnostic Tool	Treatment
Lactose	Hydrogen breath test with lactose	Low-lactose diet
Fructose	Hydrogen breath test with fructose	Low-fructose diet
FODMAPs	Elimination diet, OFC, hydrogen breath test	Low-FODMAPs diet
NCGS	Exclusion of celiac disease and wheat allergy, the Salerno Experts’ recommendations	Gluten-free/Wheat free diet/ Low-FODMAPs

FODMAPs—fermentable oligo-, di-, monosaccharides and polyols, NCGS—non-celiac gluten sensitivity, OFC—oral food challenge.

## Data Availability

Not applicable.

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
