# Peer review of "Management of Adult Patients with Gastrointestinal Symptoms from Food Hypersensitivity—Narrative Review"

_jcm, 2022, doi:10.3390/jcm11247326_

Round 1

Reviewer 1 Report (New Reviewer)

The authors made a comprehensive review regarding the management of adult patients with gastrointestinal symptoms from food hypersensitivity. The present manuscript is well-written. There are several concerns that I would like to raise.

1. Please use the American English instead of the British English (i.e. faeces, oedema).

2. The paragraph between lines 75 and 85 is not appropriate for the Introduction section. This paragraph is about “Approach to Patients”.

3. The numbering of subheadings is wrong (line 93). Please correct.

4. Line 282 and 283, 368 and 369, 373 and 375, 406 and 408, 485 and 487: The number of the table in the text does not match the actual table number.

5. Table 1 seems to be cut in the middle. Please show the entire table.

6. Acronyms/Abbreviations should be defined the first time they appear in the first table. Please define these in Table 3, 4, 5, 6, and 7.

7. Line 386: Please describe what two systems are for the readers’ understanding.

8. Line 430: When undigested lactose comes in contact with the intestinal microbiota….

Is this sentence connected to the preceding sentence? Please clarify.

9. Line 487: Table 5 (actually table 7 is correct) is not mentioned in the main text.

10. Line 510: Please explain RPAH diet in more detail.

Author Response

We would like to thank for the helpful comments. We agree with all of them and changed the manuscript according to the suggestions. Changes are shown in blue color. We have added method section and rewrote abstract as the second Reviewer  suggested. We hope the revisions will enable to accept the revised version of  manuscript.

yours sincerely 

A.Kanikowska and co-authors

Reviewer 2 Report (New Reviewer)

I am a co-author of many articles in the field of gastroenterology, but I am not a clinician. I read the manuscript with great interest. I think the issue is important and awareness of the issue should be raised.

In my opinion, the abstract does not represent the article well and it is not clear what the difference is between the introduction and the rest of the manuscript.

I believe the manuscript summarizes the subject comprehensively. However, I cannot evaluate the manuscript without presentation of a detailed methods section.

In my opinion, the article must include a detailed methods section as is acceptable according to the PRISMA 2020 checklist.

Author Response

We would like to thank for the helpful comments.

We have added method section and rewrote abstract. Changes are shown in blue color. We have written all the information about the articles' search methods with the consideration of PRISMA guidelines but being aware it may not be satisfying enough we would like to add that this article was not prepared as systematic review but more as an overview or an update on the current  knowledge and concepts concerning food hypersensitivity in adult patients with gastrointestinal symptoms.

We hope the revisions will enable to accept the revised version of  manuscript.

yours sincerely 

A.Kanikowska and co-authors

Round 2

Reviewer 2 Report (New Reviewer)

I would like to thank you for adding the methods section. However, as you mentioned, the manuscript was not written according to the principles of a systematic literature review and is therefore methodologically exposed to biases. I believe that publishing the manuscript as a review can mislead the reader.

I suggest changing the title so that it clearly indicates that this is not a review manuscript and adding in the discussion a paragraph showing the methodological limitations of not using the all the steps of the systematic literature review.

Author Response

Dear Reviewer,

Thank you very much for your comments on our manuscript.

We have added a paragraph showing the methodological limitations of not using the all the steps of the systematic literature review and the reason why we did not use it is that our aim was to summarize current knowledge and highlight the most typical food hypersensitivity syndromes for clinical practice that could be helpful in diagnosis and everyday care of the patient. Systematic review is a preferable and most useful tool for analyzing well-known research area to compare studies outcomes and therefore it presents specific, well-defined data, but it would not allow us to present some less studied hypersensitivity issues, important in differential diagnosis.

We have acknowledged the limitations of our article. We've also added information about other than systamatic review  type of article - adding "narrative review" in the title in order not to mislead the reader.

We hope this is sufficient for the manuscript to be accepted for the publication in JCM

On behalf of all co-authors

Yours sincerely

Alina Kanikowska

This manuscript is a resubmission of an earlier submission. The following is a list of the peer review reports and author responses from that submission.

Round 1

Reviewer 1 Report

The review on the management of adult patients with Gastrointestinal symptoms from food allergy was comprehensive and nicely written. Significantly updated, helpful information, and well explained. 

Some minor comments will make the review better and more clear:

1- In the introduction pls include more data on the impact of exposomes on the development of food allergies such as epigenetic changes, the following papers may help (PMID: 33668787, PMID: 27662207, PMID: 31633569)

2- Line 77-79 must be moved to the treatment.

3- Line 87-102 pls involved more data on the underlying mechanisms including the role of the innate and adaptive immune system in food allergy

4- Line 118-127 pls include more data on the advantage and disadvantages of the in vitro tests.

5- As the authors involved info on the probiotics, it would be better to add a couple of sentences on the impact of microbiota and dysbiosis in food allergy development. 

6- Line 136-147 pls add some statistical data on the success of immunotherapy in food allergy management.

7-Line 157, pls add some examples of connective tissue disorders.

8- Line 159-165 it is worth including some sentences on the role of Th-2 on the pathophysiology of food allergy. 

9- Line 187: which was more sensitive as a test the sIgE or SPT and why?

10- Line 229: how the AP was diagnosed and what was the serum markers concentration, was it the serum amylase activity or what?

11- For paragraphs 4.2 and 4.3: pls add a table summarizing the diagnostic and treatment options for carbohydrates intolerance.

Author Response

Dear Reviewer,

Thank You for very constructive and helpful review. We added all the recommended information (marked in yellow). The article has been rearranged according to second Reviewer comment, that the presented data should be focusing on type of hypersensitivity mechanisms and uncertain information/without accepted guidelines should be removed or explained. We also added more research articles supporting presented information and statements.

Sincerely

Authors

Reviewer 2 Report

General Comments

The line numbering is incorrect and difficult to refer to specific text.

The research is a review of different classifications of food hypersensitivity that cause gastro-intestinal symptoms. The purpose of the review is to provide an overview for a doctor or dietician to assist in the interpretation of symptoms and guidance of therapies. The review attempts to classify the different types of biological mechanisms, broad range of symptoms, different types of triggering foods and some of the therapies available for treatment of each classification of hypersensitivity.

The review is ambitious considering the enormous overlap of symptoms (usually self-reported) across a wide range of dietary triggers and the inter-dependence of biological mechanisms. Unfortunately, the scope is too broad and generally lacks adequate scientific evidence and supportive references for many of the statements. For example, the opening statement of the Abstract is not supported in the text with any data or evidence. In general, the text is confused and confusing with content on symptoms mixed up with therapies, and offers inappropriate recommendations for practice, without supportive data. There are examples of discussing specific therapies as if there is strong clinical evidence for these therapies, eg, last para of P15.

The challenge for food hypersensitivity is defining:   the specific symptoms in an objective way (ie, free of self-reporting bias), the biological mechanisms and the link to a specific food and preferably to the food component. Currently, much of the understanding in this topic area is heresay and apart from acute immunologically responses, eg, anaphylaxis, is based on self-reported symptoms and dietary intake. The clinical methodologies for diagnosis are not standardised and this limits scientific progress in the field. The review does not highlight these challenges or limitations and makes many general statements without evidence. This challenging topic requires systematic review of food hypersensitivity within criteria of clinically defined symptoms, and links with defined food component triggers (if possible). The review should remove all relationships to food, unless clinical evidence is presented and focus on the classification of the different mechanisms of hypersensitivity. Of great value would be the identification of specific diagnostic measures that are selective for specific biological mechanisms and are correlated with self-reported symptoms.

Specific Comments

P9: all spoiled foods can contain histamine so the list on P9 is misleading. What is high? What is low? What is the cut-off for reaction? This is the type of information that creates consumer anxiety and unnecessary food avoidance and does not belong in a scientific journal.

Author Response

Dear Reviewer,

Thank You for the review and important remarks.

Article has been changed according to Reviewer recommendations, data has been presented focusing on classification on different mechanisms of hypersensitivity, diseases without standardized diagnostic methods were clearly detached from the officially accepted ones, abbreviated to the most important information, but authors decided to present them since there has been growing number of scientific articles concerning these entities.( e.g. in Nutrients, also). Examples  of scientific research and review articles has been added to support presented information, where lacked.

All added information is marked in yellow.

Sincerely

Authors